# Exploring the delivery of empathic care in task-shared settings: A psychometric study in rural Pakistan

## Research Article

perinatal depression; empathy; community health workers; digital mental health; digital app

**Corresponding author:**
Ahmed Waqas;
Email: ahmed.waqas@liverpool.ac.uk

Rakhshanda Liaquat[1], Ahmed Waqas[2,3,4] , Tayyaba Qadeer[1], Abid Malik[5], Najia Atif[1], Siham Sikander[2,3] , Duolao Wang[6] and Atif Rahman[2] 

[1]Human Development Research Foundation, Rawalpindi, Pakistan; [2]Institute of Population Health, University of Liverpool, Liverpool, UK; [3]Mersey Care NHS Foundation Trust, Liverpool, UK; [4]Greater Manchester Mental Health NHS Foundation Trust, Salford, UK; [5]Health Services Academy, Islamabad, Pakistan and [6]Liverpool School of Tropical Medicine, Liverpool, UK

## Abstract

Empathy plays a crucial role in psychosocial and psychological interventions, greatly impacting rapport building, patient adherence, and satisfaction with treatment. Empathetic interactions enhance patient's self-reflection and the delivery of more personalized therapeutic interventions tailored to the unique needs of each patient, thereby improving the overall quality of care. Despite empathy being central to psychosocial interventions, there are currently no valid and reliable patient-centered tools that assess the lay-therapist empathy that they show and/or exhibit toward their patients.

In this study, the patient-rated Empathy Scale for Lay Therapists was developed to assess empathy in community health workers delivering psychosocial interventions. Psychometric validation was based on a cross-sectional study embedded in a non-inferiority cluster randomized trial of the Thinking Healthy Programme for perinatal depression in Pakistan.

Community testing with perinatal women confirmed the scale's understandability and logical structure, highlighting its face validity. Among the 980 trial participants, a high level of agreement with the Empathy Scale for Lay Therapists (mean score 2.616) was observed, indicating effective communication and empathy from health workers. The scale demonstrated excellent internal consistency (Cronbach's alpha 0.96). Exploratory Factor Analysis revealed a unidimensional structure, capturing 87.81% of the total variance, with strong factor loadings.

## Impact Statement

The delivery of empathic care is fundamental to effective psychosocial interventions, enhancing therapeutic rapport, patient satisfaction, and adherence to treatment. This study introduces the Empathy Scale for Lay Therapists (ESLT), a novel, patient-rated tool specifically designed to measure empathy in task-shared settings, where non-specialists deliver care. Developed and psychometrically validated within the context of a large-scale trial in rural Pakistan, the ESLT represents a significant advancement in the evaluation of empathic care in low-resource settings. Its unidimensional structure, high internal consistency, and robust psychometric properties provide a reliable framework for assessing empathy in community health workers.

The ESLT's broader impact lies in its potential to improve the quality of task-shared mental health interventions globally. It provides an evidence-based method for evaluating and enhancing empathic communication, a key determinant of treatment outcomes. In practice, the ESLT can be integrated into training programs for lay therapists, offering actionable feedback to strengthen empathic skills and improve patient-centered care. It also serves as a monitoring tool, ensuring that empathy remains a cornerstone of care delivery in peer-support and community health programs.

By facilitating research on empathy's role in improving patient outcomes, the ESLT paves the way for innovations in training, supervision, and program design. Cross-cultural validation studies will further expand its applicability, while its integration into routine performance assessments will promote sustainable improvements in care quality. The ESLT aims to empower healthcare systems, particularly in low-resource settings, to foster empathic interactions that resonate with patients' needs, enhancing the impact of task-shared interventions on mental health and well-being.

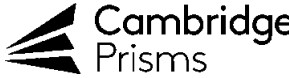

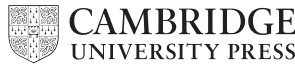

## Introduction

Task sharing has been increasingly recognized as an innovative solution to address the pervasive treatment gaps in mental health care, particularly in regions with limited resources. The concept (Rahman et al., 2013) proposes the delegation of specific therapeutic tasks to trained non-specialist health workers, under the supervision and guidance of mental health specialists. This approach not only optimizes the available workforce but also ensures the provision of mental health services that are both accessible and culturally congruent with the local population's needs. The task-sharing model is predicated on the effective distribution of care responsibilities, thereby alleviating the burden on specialized mental health professionals and facilitating a wider reach of mental health services. Recent literature has (Kakuma et al., 2011) highlighted how this model not only increases the efficiency of mental health service delivery but also enhances its relevance by incorporating interventions that are aligned with the cultural and linguistic context of the community served.

A pivotal aspect of the task-sharing model is the reliance on lay health workers, whose personal attributes significantly influence the efficacy and acceptability of the delivered psychological interventions for common mental disorders (CMDs). Our previous research (Atif et al., 2019) underscores the importance of characteristics such as empathy, trustworthiness, and shared linguistic and cultural backgrounds, which enhance the therapeutic relationship and facilitate a deeper understanding and connection between the health worker and the patient. Effective communication skills are deemed indispensable in the context of task sharing, serving as the foundation for successful patient interactions and interventions. Hemmerdinger et al. (2007) assert the critical nature of these skills in healthcare, emphasizing their role in accurately identifying and addressing patient needs. Central to effective communication is the concept of empathy, which involves a deep understanding of the patient's perspective, experiences, and emotions (Refaat Ahmed and Shalaby, 2022). Empathy is described as "putting oneself in the patient's shoes," highlighting its significance in building a genuine rapport between healthcare providers and patients, thereby augmenting the impact of task-shared interventions.

Empathy is a pivotal component within psychosocial and psychological interventions, significantly influencing patient motivation, adherence to, and contentment with prescribed treatment regimens (Williams et al., 2014). Such empathetic engagement not only heightens patient satisfaction, as noted by Berhan and Berhan (2014) but also elevates the caliber of care, fosters a deep provider-recipient connection which diminishes the likelihood of errors, and ensures the provision of therapies that are optimally tailored to individual patient needs (de Andrade Alvarenga et al., 2021). Furthermore, the presence of empathy fosters a nurturing environment, imbuing patients with feelings of security, encouragement, and confidence, thereby nurturing a foundation of trust between lay health workers and their patients. Beyond its direct benefits, empathy-driven communication catalyzes patient involvement and promotes informed decision-making, ultimately enhancing health outcomes, particularly notable in maternal health contexts (Moloney and Gair, 2015).

The evaluation of empathy within therapeutic settings is paramount for determining the quality of interactions between patients and healthcare providers, highlighting its crucial function in the therapeutic journey. Given its importance, various instruments have been devised to quantify empathy in healthcare environments.

Among these, the Jefferson Scale of Physician Empathy stands out, featuring dimensions such as perspective-taking, compassionate care, and standing in the patient's shoes, and is frequently applied in medical education research (Hojat et al., 2002). Similarly, the Empathy Quotient, utilized to gauge empathy among medical students, assesses cognitive empathy, emotional responsiveness, and social skills (Lawrence et al., 2004). Additionally, the Schwartz Center Compassionate Scale is employed to appraise the extent of compassionate care delivered by physicians and other healthcare providers (Lown et al., 2015).

Despite these advancements, a notable research gap persists in the absence of a specialized scale for measuring empathy among allied health personnel or community health workers, particularly in low- and middle-income countries. While current practices in assessing competency among such workers, especially those involved in task-shared psychological interventions, include supervision and competency exercises through role-playing, these measures do not adequately capture empathic care from the patient's perspective. For instance, the ENACT rating scale (Kohrt et al., 2015) includes only items on empathy, rated during supervision sessions. Furthermore, the ENACT tool is primarily supervisor-rated, limiting its ability to fully reflect the patient's experience of empathy during care. This highlights the need for additional tools that can accurately capture perceived empathy from the patient's perspective, particularly in culturally diverse and resource-limited settings. Developing such patient-centered empathy measures would be essential for enhancing the quality of care and ensuring that interventions are truly responsive to patient needs.

This research gap underscores the need for a novel assessment tool tailored to community health workers that accurately reflects the compassionate and empathic care they provide, as perceived by the patients themselves. Addressing this gap, our investigation is dedicated to detailing the methodologies employed in the creation of an instrument aimed at evaluating the compassionate and empathic care dispensed by community health workers, thereby contributing significantly to the field by enhancing the understanding and measurement of empathy in diverse healthcare settings.

## Methods

### The empathy scale for lay therapists

The Empathy Scale for lay therapists (ESLT) was developed to assess compassionate/empathic care delivered by community health workers (Supplementary Table 1). Unlike ENACT, which assesses competency in delivering psychosocial interventions through observed role-plays, the ESLT evaluates the level of empathic care as experienced and perceived by patients. The development process encompassed several phases. In Phase 1, a comprehensive literature review on PubMed (using keywords: empath* OR compassion* AND health-care AND scale*) was conducted to identify existing scales related to compassionate care and empathic traits among healthcare professionals. Based on this review, a committee of three experts specializing in either psychiatry or psychology and perinatal mental health research generated a long list of items aligned with the five fundamental characteristics of compassionate care. These items were translated from English to Urdu, resulting in a pool of 14 items for further review. In Phase 2, an expert panel consisting of in-house perinatal mental health and task-shared intervention experts was arranged to finalize the items through consensus. Phase 3 involved field testing of the measures in a community setting, employing face validity

procedures with key informants and focus groups of depressed mothers. Finally, in Phase 4, psychometric validation procedures were applied to a dataset generated from the 3rd-month assessments of mothers in intervention and control groups.

### Phase 1: Literature review

In this step, we reviewed previously available scales measuring either the delivery of compassionate care or empathic traits among healthcare professionals. Most notably, we reviewed the scales listed in Table 1.

A long list of items representing constructs central to empathic and compassionate care (Figure 1) was long-listed by a committee of three experts in depression and questionnaire development. The choice of these items was guided by the five fundamental characteristics of compassionate care (Lown et al., 2015; La Monica, 1981; Lawrence et al., 2004; Moudatsou et al., 2020; Rodriguez and Lown, 2019; Kohrt et al., 2015):

i. Interpersonal relationships based on empathy and emotional support
ii. Efforts to understand and relieve patients' sadness and pain
iii. Effective communication and enabling the patients' and families' participation in decisions
iv. Considering patients as persons and respecting them
v. Emphasis on holism rather than reductionism

As per Mapi research trust guidelines, these *loan* items were translated from English to Urdu language, after a two-step process (McKown et al., 2020). In this two-step process, the items were forward and back-translated by the team, to ensure semantic equivalence and cross-cultural face validity. The wording of these items was then adapted for use in task-shifted interventions, delivered either by community health workers or peers. This resulted in a pool of 14 items, for further review and shortlisting.

### Phase 2. Expert panel consensus

As the next steps, these 14 items were shared with a team of experts ($n = 6$) in perinatal mental health and task-shared interventions. At this stage, the number of items was shortlisted, and any changes advised to each of the statements were pre-finalized. The shortlisting process was guided by the criteria outlined in Figure 1, informed by the field and clinical expertise of the experts, ensuring the items reflected cross-cultural concepts of empathic care. The pre-finalized items were then field tested, to request feedback from mothers comparable to our future study sample as well as the assessment team.

### Phase 3. Field testing

To ensure face validity, the research team pilot-tested the measures in a community setting with 10 perinatal women. Key informants from the assessment and field teams conducted focus group interviews with depressed mothers, during which the Urdu translation of the ESLT was administered. Feedback was collected to confirm that the terminology used in the scale was easily understandable and culturally appropriate.

### Phase 4. Psychometric validation study design

This cross-sectional study was embedded within the stratified cluster randomized controlled trial known as the Enhanced

**Table 1.** Review of scales used for meaning empathy in health research

| Scale name | Description | Context |
|---|---|---|
| Schwartz Center Compassionate Care Scale | A 12-item scale designed to measure the extent to which patients perceive that their healthcare provider demonstrates compassion (Rodriguez and Lown, 2019) | Developed in the US to assess levels of compassion in hospital physicians |
| Syrian Empathy Scale | A 12-item scale was developed to assess levels of empathy specifically in the context of interactions between Syrian refugees and healthcare providers (Dashash and Boubou, 2021) | Syrian refugees receiving healthcare. |
| Jefferson Scale of Empathy | A 20-item scale widely used scale designed to measure empathy in the context of medical education and healthcare practice (Ward et al., 2009) | Primarily used for physicians and medical students |
| Agnew Relationship Measure | A 5-item scale measure is used to assess the quality of relationships between the therapist and client (Agnew-Davies et al., 1998) | Rates therapeutic alliance between the therapist and client |
| Counselor Rating Form | A 50-question tool used to assess various dimensions of counseling sessions, including empathy, rapport, and effectiveness (Corrigan and Schmidt, 1983) | Rates the client's perception of the counselor's appeal, expertise, and reliability. |
| Empathy Construct Rating Scale | An 84-item scale was used to evaluate empathy across different contexts, including healthcare, education, and interpersonal relationships (La Monica, 1981) | Primarily used in nursing research, this tool allows nurses to self-assess the extent to which they perceive themselves as empathic. |
| Enhancing Assessment of Common Therapeutic Factors (ENACT) | A 15-item scale focusing on assessing common therapeutic factors, including empathic care, active listening, and non-verbal communication skills (Kohrt et al., 2015) | For rating competency of non-specialist mental health professionals during supervised sessions. |

Technology-Assisted version of the Thinking Healthy Programme (THP-TA), with stratification based on the smallest district administrative unit, the Union Council. The study was specifically conducted in the sub-districts of Kallar Syedan, Gujar Khan, and Rawalpindi, representing rural areas within the Rawalpindi district.

Pregnant women aged 18 years and over, in the second to early third trimester of pregnancy (4–8 months gestation), experiencing a current Major Depressive Episode (MDE) on SCID, and intending to reside in the study area for approximately 1 year, were included. Eligibility was determined by checking the registers of Lady Health Workers (LHWs), government-employed community health workers, responsible for around 250 households each, and maintaining a register of every new pregnancy within their catchment area. Further details of the trial design can be found in the study protocol published elsewhere (Rahman et al., 2023). Details on the

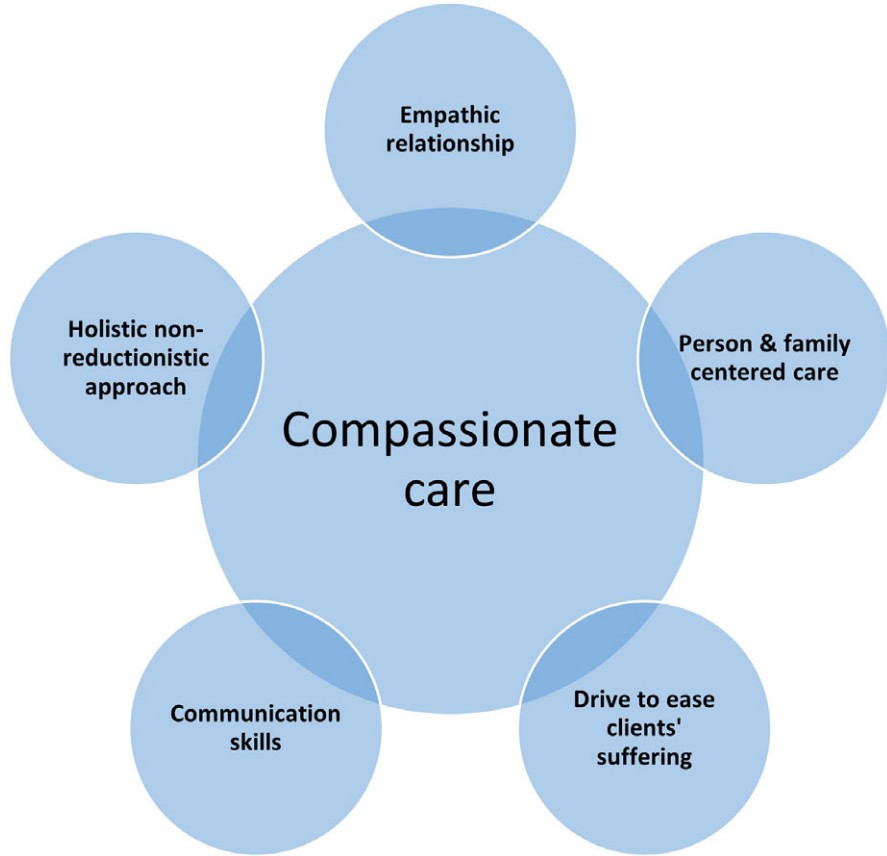

**Figure 1.** Five guiding principles for formulating questionnaire items for Empathy Scale for Lay Therapists.

development procedures of the intervention were published by Atif and colleagues elsewhere (Atif et al., 2019).

**Interview procedures:** The study's procedures were carried out as per the Declaration of Helsinki 2013 (World Medical Association, 2013). Ethical approval was obtained from the Institutional Review Board (IRB) of the University of Liverpool and the Ethics Committee at the Human Development Research Foundation, Pakistan. All participants provided informed written consent at the time of recruitment. All participants volunteered for the study and provided written informed consent. They were guaranteed anonymity and confidentiality, with the assurance that only collective findings would be reported.

Thereafter, a comprehensive questionnaire was administered by a team of research assistants. These assistants, supervised by an experienced psychiatrist, were trained in administering psychosocial instruments, obtaining informed consent, and recording participant reactions and responses through structured sessions and interactive workshops. The questionnaires were administered orally, with responses recorded on tablets using the Open Data Kit (ODK), an online data collection kit.

To establish the construct validity of the ESLT, we assessed its convergent validity by examining correlations with established measures of anxiety (GAD-7) and depression (PHQ-9) among intervention recipients. A significant positive correlation between ESLT scores and these measures would indicate that higher therapist empathy aligns with clients' mental health outcomes, as theoretically expected. In the absence of more closely related constructs, such as prosocial behavior, apathy, or emotional intelligence, this approach serves as a proxy for validating the ESLT.

While these alternative constructs might have provided a more direct validation framework, the use of GAD-7 and PHQ-9, with their established psychometric properties, provides meaningful evidence that the ESLT effectively captures a construct closely related to psychological distress, reinforcing its validity as a measure (de Andrade Alvarenga et al., 2021). For concurrent validity, it was hypothesized that ESLT would correlate positively with competency assessments during supervision sessions, using the ENACT tool.

All these measures were used in baseline, 3rd month follow-up, and 6th month follow-up assessments in the trial (Rahman et al., 2023).

*Patient Health Questionnaire 9-Itmes (PHQ-9)*: PHQ-9 is the nine-item DSM-IV symptom-based criteria for depression on a four-point Likert scale from not having the symptom at all, to having it nearly every day, over the last 2 weeks. The score for each item is summed to arrive at a total score. This screening tool has been validated and has a high positive predictive value for the diagnosis of depressive disorder and has been used extensively in Pakistan (Sikander et al., 2019).

*Generalized Anxiety Disorder 7-Items (GAD-7)*: Based on DSM-IV symptom-based criteria for generalized anxiety disorder on a four-point Likert scale from not having the symptom at all, to having it nearly every day, over the last 2 weeks. The score for each item is summed to arrive at a total score. This tool also has been extensively used in study settings (Sikander et al., 2019).

*Multidimensional Scale of Perceived Social Support (MSPSS)*: The MSPSS was used to assess the perceived levels of social support among the participants. There are 12 items and three subscales

related to social support from family, friends, and significant others. This scale has been translated and validated for use in this study setting (Sharif et al., 2021).

*The Enhancing Assessment of Common Therapeutic Factors (ENACT) Tool*: To assess peer competency in implementing the THP-TA intervention, evaluations were conducted at three distinct points: immediately post-training, 6 months after training, and 12 months post-training. These assessments utilized standardized role-play scenarios developed by the World Health Organization's EQUIP (Ensuring Quality in Psychological Support) platform, designed specifically to enhance training and supervision processes in mental health and psychosocial support services. Trained assessors, proficient in both the TA-THP intervention and the assessment instruments, employed culturally tailored role-play scripts appropriate for the Pakistani context. Each role-play session, lasting approximately 30 min, featured an actor portraying a mother with depression, with assessors who were knowledgeable in TA-THP, observing a peer's intervention delivery.

The ENACT tool (Kohrt et al., 2020) measures 15 core therapeutic domains, including skills in verbal and non-verbal communication, rapport building, empathy, harm assessment, appropriate family engagement, collaborative goal setting, psychoeducation, and eliciting feedback. Competency in each domain on both tools was rated using a four-point Likert scale, with Level 1 indicating the presence of unhelpful behaviors, while Level 4 denoted mastery of essential and advanced therapeutic competencies.

## Statistical analysis

All statistical tests were conducted employing the Statistical Package for the Social Sciences (SPSS), version 27. Descriptive statistics were used to examine participant characteristics. Mean scores, standard Deviations, and frequencies for each item of the Empathy Scale for lay therapists were also calculated. Cronbach's alpha was used to examine if internal reliability was satisfactory for further validation. We used a value of >0.7 as a cutoff for internal reliability (Streiner and Kottner, 2014). The Kaiser–Meyer–Olkin (KMO) measure was used to evaluate the adequacy of the sample for factor analysis (Kaiser, 1974). Based on a minimum desired value of 0.6, we used Bartlett's test of sphericity (Haitovsky, 1969) to examine whether the items had enough in common to justify conducting a factor analysis. We used exploratory factor analysis (EFA) to assess the factor structure of the Empathy Scale for lay therapists. Cattell's scree plot was used to determine the maximum number of components to retain in the EFA.

After the identification of an appropriate factor structure of the Empathy Scale for lay therapists, Confirmatory factor analysis (CFA) was carried out to verify the factor structure. We calculated the Root Mean Square of Residuals (RMSR), using a cutoff of <0.10 to indicate a good fit (Hu and Bentler, 1999). For the Comparative Fit Index (CFI) and Tucker–Lewis index (TLI), we used a cut-off of >0.90 (Hooper et al., 2008). Finally, we used the Goodness-of-Fit Index (GFI) and the Adjusted Goodness-of-Fit Index (AGFI). The GFI and AGFI range from 0 to 1, and >0.9 indicates an acceptable model fit (Babyak and Green, 2010).

## Results

### Face validation

The face validation phase encompassed two critical stages: expert panel review and community-based field testing. Feedback from these stages was predominantly constructive. In the expert review phase, the panel proposed three key revisions. The first recommendation was to establish a clear recall period that spanned the entire duration of the intervention sessions. The second was a terminological change, suggesting the replacement of the term "Aitemad" (confidence) with "Aitabaar" (trust) in the third item. Additionally, it was advised to avoid compound concepts, such as combining 'attention' and 'concentration' in item six, opting instead to solely use "tawajja" (attention). The expert team also recommended using a 4-point Likert scale to assess respondents' levels of agreement with statements. Participants indicate their responses on a scale ranging from 0 to 3, where 0 represents "Strongly Disagree, 1 represents "Disagree," 2 represents "Agree," and 3 represents "Strongly Agree."

Field testing with perinatal women further affirmed the instrument's relevance and comprehensibility. Participants uniformly concurred that the scale was logical and easy to understand, thereby underscoring its face validity in the target population.

### Characteristics of the study sample

A total of 2,861 women were approached for participation in the study, out of which 99 were excluded due to not meeting level 1 exclusion criteria. Out of the 2,760 remaining who agreed to participate and fulfilled the inclusion and exclusion criteria, they were assessed for depression using the Structured Clinical Interview for DSM-IV. Out of 2,760 participants, 980 (35.51%) fulfilled the diagnostic criteria for depression and were recruited for the study. Participants in both the control and intervention arms received therapy administered by therapists. The therapy was delivered through therapist-led sessions. The mean age of the sample was 29.31 years (Table 2).

### Response distribution on individual statements

The mean score for ESLT responses was approximately 2.616 (0.5092), indicating a generally high level of agreement (Supplementary Figure 1). Specifically, when participants were asked if the lay therapist communicated in a manner that was easy to understand, 530 (54.08%) participants strongly agreed with this statement, while only 2 participants strongly disagreed with item 10, whether the lay therapist acknowledged the possibility of experiencing emotions in difficult situations (Table 3, Supplementary Table 2).

### Reliability

The reliability of the ESLT scale was assessed using Cronbach's alpha coefficient, a measure of internal consistency reliability. The scale demonstrated high levels of internal consistency, as indicated by a Cronbach's alpha coefficient of 0.96. The range of Cronbach's alpha coefficients if items are deleted ranges from 0.962 to 0.966 (Supplementary Tables 3 and 4).

### Exploratory factor analysis (EFA)

Principal axis factoring was undertaken to explore latent constructs and to assess the dimensionality of this exploratory factor (Table 4). The KMO measure of sampling adequacy yielded an impressive value of 0.950, indicating that the dataset was highly suitable for factor analysis. Additionally, Bartlett's test of sphericity returned a statistically significant result ($p < 0.05$), providing further evidence of

**Table 2.** Demographic characteristics of study participants (*n* = 980)

| | | Mean | SD | Frequency | % |
|---|---|---|---|---|---|
| Age | | 27 | 5 | | |
| Duration of pregnancy in weeks | | 21 | 5 | | |
| Participants' education | No schooling | | | 87 | 8.9% |
| | Primary school | | | 108 | 11.0% |
| | Middle school | | | 195 | 19.9% |
| | Matric or higher | | | 590 | 60.20% |
| Structure of family | Nuclear | | | 215 | 21.9% |
| | Joint | | | 367 | 37.4% |
| | Extended | | | 327 | 33.4% |
| | Multiple households | | | 71 | 7.2% |
| How many people live in your home | | 8 | 4 | | |
| Participant's employment status | Not employed | | | 952 | 97.1% |
| | Employed | | | 28 | 2.9% |
| Estimated monthly income | | 32,028 | 48,653 | | |
| Is it your 1st pregnancy | No | | | 758 | 77.3% |
| | Yes | | | 222 | 22.7% |
| Do you live with your husband? | No | | | 92 | 9.4% |
| | Yes | | | 888 | 90.6% |

**Table 3.** Total response count for the empathy scale

| Scale items | Strongly disagree (*n*, %) | Disagree (*n*, %) | Agree (*n*, %) | Strongly agree (*n*, %) |
|---|---|---|---|---|
| ESLT11: Did the lay therapist explore your expectations with this program? | 6 (0.61%) | 7 (0.71%) | 311 (31.73%) | 495 (50.51%) |
| ESLT10: Did the lay therapist tell you that in difficult situations such emotions can be experienced? | 2 (0.20%) | 8 (0.82%) | 314 (32.04%) | 495 (50.51%) |
| ESLT9: Did your lay therapist try to understand your health problems and their impact on you? | 4 (0.41%) | 9 (0.92%) | 301 (30.71%) | 505 (51.53%) |
| ESLT8: Did the lay therapist summarize what you had said to help you know that she understood you? | 3 (0.31%) | 6 (0.61%) | 311 (31.73%) | 499 (50.92%) |
| ESLT7: Did the lay therapist acknowledge your feelings? | 4 (0.41%) | 6 (0.61%) | 299 (30.51%) | 510 (52.04%) |
| ESLT6: Did the lay therapist use open-ended questions? | 4 (0.41%) | 10 (1.02%) | 293 (29.90%) | 512 (52.24%) |
| ESLT5: Did you feel that your lay therapist listened to you with full attention (e.g., through her eye contact, body language, and posture)? | 4 (0.41%) | 4 (0.41%) | 290 (29.59%) | 521 (53.16%) |
| ESLT4: Did you feel that you can trust your lay therapist? | 6 (0.61%) | 10 (1.02%) | 269 (27.45%) | 534 (54.49%) |
| ESLT3: Did you feel that your lay therapist was compassionate towards you? | 5 (0.51%) | 7 (0.71%) | 273 (27.86%) | 534 (54.49%) |
| ESLT2: Did the lay therapist show respect to you? | 3 (0.31%) | 3 (0.31%) | 271 (27.65%) | 542 (55.31%) |
| ESLT1: Did the lay therapist talk to you in a way that was easy to understand? | 4 (0.41%) | 2 (0.20%) | 283 (28.88%) | 530 (54.08%) |

the appropriateness of the data for factor analysis. Utilizing the criteria for Eigenvalues >1 and Cattell's scree plot (Supplementary Figure 2), only one factor was retained. The scree plot and eigenvalues derived from the factor analysis revealed a clear and discernible unidimensional factor structure within the dataset. The factor analysis elucidated a total variance of 87.81% in Empathy scale scores. Importantly, all items demonstrated adequate estimates for commonalities, with values ranging from 0.6 to 0.8. All items yielded strong factor loadings ranging from 0.63 (item 7) to 0.81 (item 11).

### *Confirmatory factor analysis*

CFA was used to further test the goodness of fit of the unidimensional factor structure for the ESLT (Figure 2). All 12 items yielded strong factor loadings ranging from 0.76 (items 1 and 11) to 0.91 (item 7). However, it yielded poor goodness of fit indices (CFI = 0.86, GFI = 0.71, AGFI = 0.058, RMR = 0.16, RMSEA = 0.19, and CMIN/DF = 29.91). An analysis of modification indices suggested covarying residual errors between items 1 and 2, items 3 and, items 7 and 8, items 9 and 10, and items 11 and 12. These modification indices were done serially. The final unidimensional model yielded acceptable goodness of fit indices (CFI = 0.94, NFI = 0.94, GFI = 0.87, AGFI = 0.80, PCMIN/DF = 14.12, RMSEA 0.13, and RMR = 0.01).

### *Convergent validity*

The correlation between the ESLT and PHQ-9 scores was statistically significant but negative ($r = -0.369$, $p < 0.001$), indicating that higher levels of empathy, as measured by the ESLT, are associated with lower levels of depression symptoms. Similarly, the correlation between the ESLT and GAD-7 scores was also statistically significant but negative ($r = -0.250$, $p < 0.001$), suggesting that higher levels of empathy are associated with lower levels of anxiety

**Table 4.** Exploratory factor analysis for empathy scale for lay therapist communalities and factor loadings of one-factor model

| Statements | Communalities | Factor loading |
|---|---|---|
| ESLT1: Did the lay therapist talk to you in a way that was easy to understand? | 0.800 | 0.640 |
| ESLT2: Did the lay therapist show respect to you? | 0.830 | 0.688 |
| ESLT3: Did you feel that your lay therapist was compassionate towards you? | 0.849 | 0.722 |
| ESLT4: Did you feel that you can trust your lay therapist? | 0.865 | 0.748 |
| ESLT5: Did you feel that your lay therapist listened to you with full attention (e.g., through her eye contact, body language, and posture)? | 0.890 | 0.792 |
| ESLT6: Did the lay therapist use open-ended questions? | 0.891 | 0.793 |
| ESLT7: Did the lay therapist acknowledge your feelings? | 0.901 | 0.813 |
| ESLT8: Did the lay therapist summarize what you had said to help you know that she understood you? | 0.890 | 0.792 |
| ESLT9: Did your lay therapist try to understand your health problems and their impact on you? | 0.865 | 0.749 |
| ESLT10: Did the lay therapist tell you that in difficult situations such emotions can be experienced? | 0.865 | 0.748 |
| ESLT11: Did the lay therapist explore your expectations with this program? | 0.791 | 0.625 |
| ESLT12: Did your lay therapist build your hope that your health will get better by participating in the programme? | 0.820 | 0.672 |

symptoms. The correlation between the ESLT and MSSPS scores was found to be statistically significant ($r = 0.423$, $p < 0.001$). This positive correlation suggests that higher scores on the ESLT are associated with higher perceived social support, supporting the convergent validity of the ESLT as a measure of empathy.

### *Concurrent validity*

Using ENACT, the initial assessment for competency in the delivery of the THP, took place the day after training concluded, with all peers achieving a minimum of Level 2 across all domains, indicating that none displayed harmful behaviors immediately following training. As the delivery agents accumulated experience, their scores improved. By the 6-month post-training evaluation, most delivery agents had advanced to Level 3, reflecting proficiency in fundamental skills. At the 12-month assessment, the majority had either sustained Level 3 or progressed to Level 4, indicating mastery of advanced skills (Supplementary Figure 3).

The ESLT did not yield statistically significant correlations with total ENACT scores, assessed during supervision sessions at post-intervention ($r = -0.14$, $p = 090$) or long-term follow-up ($r = -1.70$, $p = 0.13$).

## Discussion

The ESLT scale demonstrated robust psychometric properties. The principal axis factoring analysis confirmed a unidimensional factor structure, accounting for 87.81% of the total variance. CFA initially produced suboptimal goodness of fit indices; however, after addressing residual correlations, the revised model showed satisfactory fit statistics. The ESLT emerges as a reliable tool for measuring lay therapists' empathic abilities.

The ESLT's effectiveness is further evidenced by its positive association with the MSSPS, suggesting that therapists with higher ESLT scores are perceived by clients as providing greater social support. This correlation aligns with existing literature, such as the work by Agnew-Davies et al. (1998), which links empathy in therapeutic contexts to enhanced social support experiences for clients. Importantly, our study, derived from trial data, indicates that clients of therapists with higher perceived empathy levels experience notable improvements in their anxiety and depression symptoms. This outcome resonates with prior research, including studies by Corrigan and Schmidt (1983) and La Monica (1981), which connect empathy with positive mental health outcomes. Therefore, the ESLT not only measures a specific dimension of empathy relevant to therapeutic settings but also highlights the crucial role of a therapist's empathy in ameliorating a client's mental health issues.

The ESLT is a new tool designed to specifically measure empathy among non-specialist providers, building on the foundation set by earlier research, especially the ENACT scale. The ENACT scale, as studied by Kohrt et al. (2015), has been proven to be a valid and reliable method for evaluating a range of therapeutic skills across various groups and settings. Our research adds to this knowledge by focusing on how empathy can be measured in lay therapists in Pakistan.

Unlike the ENACT scale, which assesses a broad range of therapeutic competencies during role-play sessions under supervision, the ESLT is dedicated solely to understanding empathy from the perspective of patients. This singular focus allows the ESLT to provide a deeper and more nuanced look at empathetic behaviors that are particularly important for lay therapists. This approach aligns with the modern understanding of empathy as a multifaceted attribute, encompassing not only emotional sharing but also compassionate care, social engagement, cognitive understanding, and the ability to recognize and respond to others' feelings (Moudatsou et al., 2020). By adopting this comprehensive perspective, the ESLT offers a more detailed and multidimensional evaluation of empathy, capturing its various aspects beyond mere emotional connection. Thus, by focusing on patient perceptions, the ESLT provides critical insights into the empathetic dynamics that unfold during actual therapy sessions. This highlights the complementary nature of the two tools – ENACT gauges technical competencies, while the ESLT offers a lens into the relational and empathetic elements of care, providing a holistic understanding of therapeutic effectiveness. The language used in ESLT is jargon-free and easily understandable by recipients, even those with low literacy. For example, technical terms like 'normalization' are replaced with simpler explanations, such as 'in difficult situations, such emotions can be experienced'. Furthermore, the ESLT is conducted in real-world settings, where recipients actively participate in sessions and provide feedback on their perceived empathy. In contrast, demonstrating empathy in a role-play setting, as assessed by ENACT, can be challenging due to its inherently artificial and potentially contrived nature, which may compromise the accuracy of the

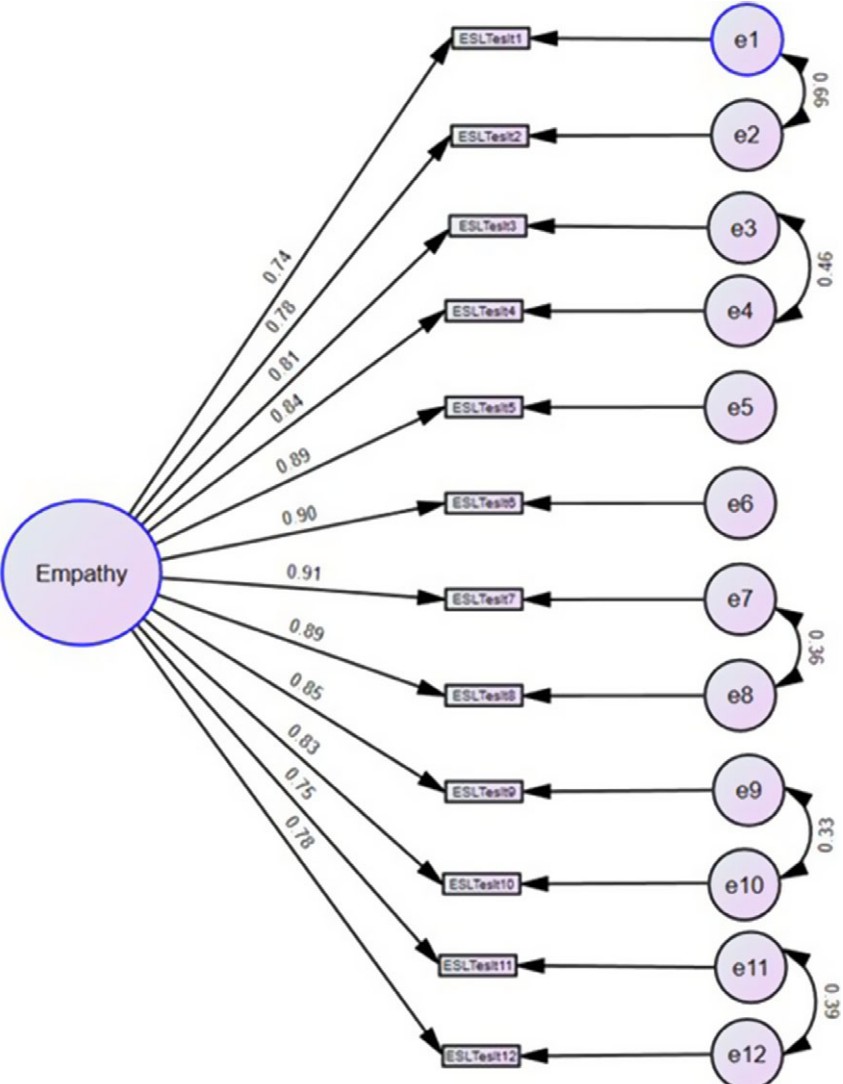

**Figure 2.** Confirmatory factor analysis for ESLT scale.

assessment. This distinction between the ESLT and the ENACT was also highlighted in our analyses where the ESLT scores did not yield statistically significant correlations.

In comparison to established empathy scales such as the Jefferson Scale of Empathy (Ward et al., 2009), the Empathy Construct Rating Scale (La Monica, 1981), the Schwartz Center Compassionate Care Scale (Rodriguez and Lown, 2019) and the Syrian Empathy Scale (Dashash and Boubou, 2021), it's noteworthy that none of the scales provided are specifically designed to measure empathy skills in lay therapists. While the literature review highlights various empathy scales tailored to specific professions like physicians and counselors, there is a noticeable gap in instruments designed explicitly for allied health workers. The absence of specialized tools for this crucial group underscores the need for further research and instrument development to address the unique empathic challenges faced by allied health professionals. It also demonstrates notable advancements in psychometric properties. The ESLT's high internal consistency reliability (α = 0.96) surpasses the reliability coefficients reported for many existing scales, indicating its robustness in measuring empathic abilities among lay therapists.

### Limitations

The study has three key limitations. First, the correlation between ENACT and ESLT scores may be confounded by factors such as assessment anxiety, which can impact therapists' performance during competency evaluations. ENACT assessments, conducted in controlled settings, may not fully capture therapists' real-world competencies or the empathy experienced by intervention recipients during actual therapy sessions. This highlights the importance of directly assessing empathy from the patient's perspective, as provided by the ESLT. Second, the study lacks broader validity analyses, particularly convergent and divergent validity assessments. While GAD-7 and PHQ-9 were used as proxies for related constructs, the absence of more relevant tools – such as those measuring prosocial behavior, apathy, or emotional intelligence – limited the depth of our validation efforts. Additionally, the lack of divergent validity testing restricts our ability to confirm the ESLT's specificity in measuring empathy without overlapping with unrelated constructs. Future research should address these gaps to enhance the robustness of the ESLT's validation. Lastly, quantifying empathy – a highly nuanced construct – based on the perceptions of

individuals receiving psychosocial interventions is inherently influenced by their mental health status. As such, these findings should be interpreted with caution.

## Recommendations for research, and practice

### Recommendations for research

Future research should focus on conducting cross-cultural validation studies of the ESLT to ensure its applicability and reliability across diverse interventions, populations, and settings. Additionally, the scale could be utilized in studies evaluating the effectiveness of empathy training programs for health workers and lay therapists by assessing pre- and post-training empathy levels. Exploring the relationship between high ESLT scores and patient outcomes – such as satisfaction, treatment adherence, and overall well-being – would help establish the scale's predictive validity. Complementing these assessments with qualitative interviews could provide deeper insights into patient experiences and perspectives on empathy in healthcare interactions, further enriching our understanding of its impact.

### Recommendations for practice

The ESLT can be integrated as a feedback tool in empathy training workshops for health workers and lay therapists, enabling them to identify strengths and areas for improvement in their empathic interactions. The ESLT scores could also inform the development of patient-centered care models that prioritize empathic communication and relationship-building. In peer support programs, the ESLT can serve as a monitoring tool to enhance the quality of empathic interactions, ensuring peers are effectively trained in these critical skills. Additionally, incorporating the ESLT into regular performance evaluations for health workers and lay therapists could encourage continuous professional development and a sustained focus on empathic practices.

## Conclusion

This study highlights the ESLT as a reliable and valid tool for measuring empathy in lay therapists from the perspective of intervention recipients. Its development addresses a critical gap in the assessment of empathy, offering a patient-centered approach that complements existing competency measures like ENACT. Future research should focus on expanding the ESLT's validation framework, including convergent and divergent validity assessments with other relevant constructs. By providing a robust measure of empathy, the ESLT holds the potential to improve the quality and effectiveness of community-based psychosocial interventions.

**Open peer review.** To view the open peer review materials for this article, please visit http://doi.org/10.1017/gmh.2025.4.

**Supplementary material.** The supplementary material for this article can be found at http://doi.org/10.1017/gmh.2025.4.

**Data availability statement.** All data are available from the corresponding author upon reasonable request.

**Author contribution.** A.R., A.W., N.A., and S.S. designed the study. R.L., A.W., and N.A. designed the scale. R.L., T.Q., and A.M. conducted the field operations and collected the data. A.W. analyzed the data and interpreted the results. R.L. and A.W. wrote the initial draft of the manuscript. A.W. revised the manuscript. All authors reviewed the manuscript and provided with critical revisions. All authors approved the final submission.

**Financial support.** This study was funded by the National Institute for Health and Care Research, United Kingdom (Research and Innovation for Global Health Transformation; Award ID: NIHR200817).

**Competing interest.** The authors do not have any competing interests to report.

**Ethics statement.** The study's procedures were carried out as per the Declaration of Helsinki 2013 (Association, 2013). Ethical approval was obtained from the Institutional Review Board (IRB) of the University of Liverpool and the Ethics Committee at the Human Development Research Foundation, Pakistan. All participants provided informed written consent at the time of recruitment. All participants volunteered for the study and provided written informed consent. They were guaranteed anonymity and confidentiality, with the assurance that only collective findings would be reported.

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
