## [Reviewer Report]

This study examines the psychometric properties of an empathy scale for lay health workers. This focus is much appreciated as there is a great need to better evaluate the suitability of individual lay health workers for task-shared mental health care as services scale. I have made some suggestions below to further strengthen this paper.

Abstract

- I don’t think it comes across clearly in the abstract why it is important to evaluate empathy (though it does in the introduction). Please try to elaborate on this here. It was also not clear to me until reading the introduction that this is a patient-driven scale so I would emphasize that point, and why it is important to have patient-centered measures in general, further in the abstract too.

Introduction

- The ENACT does include one item on empathy. I would rephrase this on page 4 to make it clearer that what you are saying is that it is not sufficient, rather than that it doesn’t evaluate it.

- I would also include a paragraph on the importance of patient (rather than provider) centered measures in your introduction. This is an important point that comes up in the discussion but I would frame the need for this measure in general around that in the intro too.

Methods

- Please describe the expertise and backgrounds of the three experts in Phase 1.

- Please describe how your comprehensive literature review was conducted.

- I don’t think the ENACT was ever 25 items.

- In the table of the other measures could you please describe briefly the context in which each measure was first developed?

- Please describe how items were shortlisted in Phase 2 - under what criteria did something make the shortlist?

- Can you please describe Phase 3 procedures to establish face validity more thoroughly? How many focus groups were conducted for example? How was face validity established?

- I am unclear on how the MSPSS would be useful for convergent validity as it assesses a different construct than your new measure - overall social support is not the same as empathy from a provider. I would have suggested using an existing measure not meant for LHWs instead. Please clarify. Likewise, I assume you mean PHQ-9 and GAD-7 are measures of divergent validity. But as symptom measures these seem completely different than perceptions of empathy. Please clarify why these are useful measures of divergent validity here.

Results

- I am struck by the fact that so few people reported strongly disagree or disagree across all items. It seems like this makes convergent validity even more important to assess (on one of the other measures you found in the literature or as assessed by someone else). Or alternatively I am wondering if there may have been some kind of response bias. Please comment on the skew here and also how it may have influenced psychometric results.

Discussion

- Is the tool meant to be used by patients in the future? Or by providers? The recommendations for practice are much appreciated but I think that particular piece of who you see using it could be made clearer.

- Please include a paragraph on limitations and then close with a concluding paragraph.

---

## [Reviewer Report]

Thank you for the opportunity to review this manuscript. The authors have developed a novel scale for the measurement of client perspectives of the level of empathic care offered by task-shared mental health providers. The approach to measure development and initial psychometric validity testing was robust, though I have a few moderate/minor concerns outlined below. The paper is well-written and concise, though there are a handful of grammatical issues throughout. I have noted some below. The proposed scale will be a useful complement to ENACT for assessing the competence and quality of care provided by task-shared caregivers.

Moderate concerns:

- Page 9, lines 3-8 – how did you choose these scales for assessment of convergent/divergent validity? These are measuring constructs that are related to the psychological and sociological status of the respondent and are not measures of their perception of the care they receive from their task-shared provider. It seems that it would be necessary to establish divergent (and convergent) validity by comparing the empathy scale scores against other (distinct and similar, respectively) measures of perceived provider care quality. For example, evidence of convergent validity would include strong correlation between this measure and other existing measures of empathic care. Evidence of divergent validity would include weak (approaching 0) correlation between this measure and other existing measures of care competence not related to empathy.

- You mention that the scale was developed in Urdu – is there an English version available? Will you provide a version of the scale as supplemental information with this manuscript, in Urdu and/or English?

- Are there any apparent cut-off scores or other opportunities to categorize scale responses to increase utility?

- You measured reliability using Cronbach’s alpha (internal consistency). In addition, it would be useful to assess reliability in terms of the correlation of client responses across individual providers – that is, do clients of the same provider agree about the level of their empathic care?

Minor concerns:

- Consider including a brief discussion or conclusion section in your abstract.

- Last sentence of second introduction paragraph (page 3, lines 40-43), revise for grammar.

- Authors note that other measures of empathy in other therapeutic settings exist (page 4, lines 10-25), but do not describe the point of measuring empathy beyond understanding the extent of compassionate care. What other uses are there for measuring empathy in clinical care? What do others do with this information?

- Page 4, lines 27-30, revise for grammar

- Page 4, lines 35-37 – again, what exactly would be the point of having patient-level perspective on empathic care? Suggest including here a brief description of what service providers/programs could do with this information (e.g., some of the points you make in the ‘recommendations for practice’ section).

- Page 5, line 5, how many items were in the expert-generated list? Also clarify on page 6, line 11.

- Page 7, line 41, spell out the MAPI acronym and include citation and description.

- Page 10, line 50, clarify that 2760 were positive for depression, and 980 (35%) were recruited into the study. Or were only 35% of the 2760 positive for depression?

- Table 2 – I’m not convinced we need this level of detail, e.g., individual rows for the number of children and for each individual occupation. I suggest simplifying and collapsing rows where you can.